# Role of Specific Autoantibodies in Neurodegenerative Diseases: Pathogenic Antibodies or Promising Biomarkers for Diagnosis

**DOI:** 10.3390/antib12040081

**Published:** 2023-12-08

**Authors:** Dimitrina Miteva, Georgi V. Vasilev, Tsvetelina Velikova

**Affiliations:** 1Department of Genetics, Faculty of Biology, Sofia University “St. Kliment Ohridski”, 8 Dragan Tzankov Str., 1164 Sofia, Bulgaria; 2Medical Faculty, Sofia University St. Kliment Ohridski, 1 Kozyak str, 1407 Sofia, Bulgaria; vvasilev.georgi@gmail.com (G.V.V.); tsvelikova@medfac.mu-sofia.bg (T.V.); 3Clinic of Neurology, Department of Emergency Medicine UMHAT “Sv. Georgi”, 4000 Plovdiv, Bulgaria

**Keywords:** neurodegenerative disorders, autoantibodies, biomarkers, pathogenesis, amyloid-β antibodies, α-Synuclein autoantibodies, Alzheimer’s disease, Parkinson’s disease, anti-myelin basic protein autoantibodies, multiple sclerosis, neurofilament autoantibodies, amyotrophic lateral sclerosis

## Abstract

Neurodegenerative diseases (NDDs) affect millions of people worldwide. They develop due to the pathological accumulation and aggregation of various misfolded proteins, axonal and synaptic loss and dysfunction, inflammation, cytoskeletal abnormalities, defects in DNA and RNA, and neuronal death. This leads to the activation of immune responses and the release of the antibodies against them. Recently, it has become clear that autoantibodies (Aabs) can contribute to demyelination, axonal loss, and brain and cognitive dysfunction. This has significantly changed the understanding of the participation of humoral autoimmunity in neurodegenerative disorders. It is crucial to understand how neuroinflammation is involved in neurodegeneration, to aid in improving the diagnostic and therapeutic value of Aabs in the future. This review aims to provide data on the immune system’s role in NDDs, the pathogenic role of some specific Aabs against molecules associated with the most common NDDs, and their potential role as biomarkers for monitoring and diagnosing NDDs. It is suggested that the autoimmune aspects of NDDs will facilitate early diagnosis and help to elucidate previously unknown aspects of the pathobiology of these diseases.

## 1. Introduction: Neurodegenerative Diseases and the Role of the Immune System

Neurodegenerative diseases (NDDs) are widely recognized as debilitating and incurable disorders with rising prevalence worldwide, especially within the elderly population [1]. These disorders can be broadly classified according to their main clinical presentation (i.e., dementia, motor neuron disease, parkinsonism), the morphological distribution of the injury (i.e., frontotemporal degenerations, extrapyramidal disorders, spinocerebellar disorders), or their primary molecular abnormality [1,2]. The most commonly encountered NDD groups include tauopathies, amyloidosis, α-synucleinopathies, proteinopathies associated with transactivation response DNA-binding protein 43 (TDP-43), etc. [2].

Although NDDs are classically associated with the accumulation of specific proteins and special neuronal population vulnerability, those disorders share many pathophysiologic processes with autoimmune and neoplastic diseases, such as alterations in the ubiquitin–proteasomal and other systems (i.e., autophagosomal/lysosomal), oxidative stress, apoptosis, and neuroinflammation [3].

In line with this, neuronal damage in NDDs often occurs in a complicated “battlefield” among resident and infiltrating immune system cells, their activated surface receptors, and secreted inflammatory modulators. The concept of the critical role the immune system plays in the processes of neurodegeneration has been recently confirmed through the identification of numerous immune factors associated with the elevated risk of NDDs in genome-wide studies [4].

Moreover, it has been postulated using research on animal models, which are genetically modified, as well as via longitudinal patient studies, that neuroinflammation and immune activation in the central nervous system (CNS) develop early in the neurodegeneration progression, most likely before large-scale neuronal loss. Microglia, representing the main CNS-resident macrophage population, are activated, secreting immunomodulatory mediators in nearly all NDDs [5,6].

The astrocyte is the second glial cell population crucial for neuroprotection and normal brain physiology. Astrocytes are multifunctional cells developed from the neuroectoderm, playing critical roles in neuronal circuitry development, synaptic pruning, and metabolism. When activated by innate immune molecules, neurotransmitters, and hypoxia, astrocytes release plenty of cell-signaling molecules. Together with microglial secretory products, these molecules mediate inflammatory mechanisms that can be helpful or harmful in NDDs [7].

A broader view of the neuropathological pathways may suggest that neuroinflammation is strictly damaging and toxic to intact residing neurons. So, we should alleviate, or at least suppress, the pathogenic process by targeting it exogenously. However, some animal studies in NDDs have demonstrated that, on the contrary, pro-inflammatory interventions may lead to better survival outcomes. These results show that there cannot be such splitting to neuroinflammation, and a more targeted modulation is favored [4,8].

As there are inflammatory mediators that cause neuronal damage and degeneration, there are circumstances that limit inflammation. Numerous harmful loop mechanisms have been observed in the CNS that decrease inflammation. Anti-inflammatory and regulatory cytokines, such as TGF-β and IL-10, are expressed along with pro-inflammatory mediators. Inhibitory molecules of inflammatory pathways, including transcriptional modulators such as activating transcription factor 3 (ATF3), suppressor of cytokine signaling (SOCS), and nuclear factor erythroid-2-related factor 2 (NRF2), can also be synthesized during the immune mechanisms involved in neuronal injury [8].

In line with this, the relationship between vitamin D and immunomodulation and homeostasis maintenance is well established. Cumulative data show that vitamin D alleviates neurodegeneration. There are many underlying pathophysiology mechanisms in NDD development where vitamin D could interfere: limiting oxidating stress and inflammation and inhibiting pathologic protein production and aggregation [9].

Therefore, understanding the immune system involvement in NDDs is a cornerstone in the potential development of immune-targeted treatment options for these disorders. At the same time, there is a consensus on the maladaptive response of neuroinflammatory mechanisms to maintain a neurotoxic environment and cause neuron lesions. However, the option is still to use recently discovered immune pathways to gain beneficial neuroinflammatory actions. A possible therapeutic target seen among these disorders includes enhancing the function of phagocytic cells to clear protein aggregates (i.e., amyloid beta (Aβ), α-synuclein). Secondly, the proper expression and modulation of reactive oxygen species (ROS) via astrocytes using the cytoprotective role of NRF2 target genes would be hypothesized to ameliorate neurotoxicity. Eventually, properly recruiting adaptive immune cells from within the CNS and the periphery could be explored as a promising strategy to promote beneficial CNS immune responses [4].

This review aims to provide data on the immune system’s role in NDDs, the pathogenic role of some specific Aabs against molecules associated with the most common NDDs, and their potential role as biomarkers for monitoring and diagnosing NDDs. It is suggested that the autoimmune aspects of NDDs will facilitate early diagnosis and help to elucidate previously unknown aspects of the pathobiology of these diseases.

## 2. Autoantibodies—Biology and Pathogenic Role in Neurodegenerative Diseases

There has been mounting evidence that there may be a potential link between Aabs and the occurrence and progression of NDDs [10]. The notion that Aabs could participate in primary NDDs is not so new. It has long been hypothesized that antibodies directed towards neuronal surface antigens might be involved in numerous NDDs, like the pathogenesis of autoimmune encephalitis. Antibodies against proteins such as amyloid β (Aβ), tau, and alpha-synuclein exist in healthy subjects and patients with NDDs. They could play a role in physiological responses to prevent accumulation or even exert additional damage [11]. The discovery that antibodies against neuronal antigens can lead to brain dysfunction has caused a paradigm shift in neurological and psychiatric disorders over the past decades while offering new diagnostic and therapeutic options. Furthermore, these findings have enabled the reclassifying of diseases previously attributed to infectious, psychogenic, or idiopathic origins [12].

Several Aabs previously associated only with acute encephalopathies have been found in patients with an insidious onset of cognitive impairment and movement disorders resembling classical NDDs. A progressive anti-LGI1 encephalitis with isolated late-onset cognitive dysfunction has been described as a treatable imitator of Alzheimer’s disease (AD) [13]. At the same time, antibodies against voltage-gated potassium channels are related to an encephalopathy originating as a frontotemporal dementia-like syndrome [14]. A clinical case in 2019 reported a confirmed post-mortem GABA_B_-receptor encephalitis in a patient previously considered to have amyotrophic lateral sclerosis (ALS) [15].

A second group of antibodies in NDDs arises secondary to neurodegeneration. It is currently not entirely established whether they are just bystanders of the degeneration or essential drivers of pathogenic mechanisms. ALS is characterized by a myriad of deposited antibodies (i.e., to LRP4) and a strong inflammatory immune response, believed to drive the process of neurodegeneration further [16,17].

A third category of antibodies renders the line between neurodegenerative and autoimmune diseases even thinner. Patients positive for anti-LON5 Aabs manifest with sleep disorders, involuntary movements, and cognitive deterioration. It was considered a typical autoimmune disorder until histopathological studies showed the hyperphosphorylated tau protein deposition as typical for neurodegeneration (in the tau-pathies). These findings support the theory that some Aabs may be the drivers of neurodegenerations, and the target interventions suppressing malfunctioning immune pathways could delay the progression of and/or even prevent NDDs [12,18].

Furthermore, low levels of “smoldering” antibodies have been observed that modulate synaptic proteins, ion channel expression, and protein aggregation. Although their role is undoubtedly pathogenic at a molecule level, when they exist as a part of a broad spectrum, they may not always cause a clinically significant disease. Thus, humoral immunity can be present or absent in neurodegeneration and slightly change the course of protein misfolding, deposition, and clearance [19].

Emerging research consistently shows that, given the mentioned effects of neuronal antibodies and the possibility of smoldering humoral immunity in the CNS, potential biomarkers for predicting NDD development would be the prevalent combination of circulating antibodies, their titers in the cerebrospinal fluid (CSF), and their duration [12].

Apart from autoantibodies, toxic protein oligomers may serve as biomarkers for neurodegenerative diseases. Oligomers are soluble protein aggregates that form intermediate structures during the aggregation of amyloid-like proteins. These oligomers, ranging in size from dimers to larger forms, have been implicated in neurodegenerative diseases [20]. Studies have shown that certain disease-related oligomers, including those associated with Alzheimer’s and prion diseases, exhibit direct concentration-dependent cytotoxic effects on neurons [21]. Some studies suggest that the presence of specific oligomers may precede the formation of mature fibrillary aggregates, potentially serving as early markers of disease onset. While having a diagnostic value even earlier in the pathologic process than antibodies, their laboratory identification poses numerous challenges because of their transient and dynamic nature. Despite these challenges, there is growing interest in identifying specific oligomeric species as biomarkers for neurodegenerative diseases. The detection of disease-specific oligomers in cerebrospinal fluid or blood may provide valuable diagnostic clues in the earliest stages of disease [22].

It is essential to recognize that not all neurodegenerative diseases may involve antibodies as a part of their pathogenesis. The research in the field of neuro-immunology is constantly evolving, and new evidence may emerge. As a result, it is challenging to definitively state that any particular neurodegenerative disease has absolutely no antibody involvement [23,24].

## 3. Autoantibodies against Pathology-Related Molecules of the Most Common Neurodegenerative Diseases

The blood–brain barrier (BBB) protects against exposure to antigens and limits the infiltration of antibodies (Abs). This is also aided by a meningeal lymphatic vessel (mLV) system, which prevents Abs from entering the CNS and directs immune cells outside to the lymph nodes [25,26]. Some NDDs are linked to BBB disintegration, which can lead to the uncontrolled leakage of neuronal and/or glial proteins and the activation of an abnormal immune response [27]. As a result, various Aabs are formed that target these proteins and contribute to brain dysfunction and neurodegeneration. Aabs associated with NDDs can be assessed in the CSF and blood (serum) of patients with NDDs. Usually, their concentration in the blood is significantly higher than in the CSF. However, Aabs could enter from the bloodstream into the nervous system in cases of a disrupted BBB or to be produced locally. In the following sections of the paper, we explored the localization of different antigens, the Aabs produced against them, and where the Aabs are located.

Still, in both cases, these Aabs can guide specific stages of the disease and/or associated comorbidities [28]. Selected Aabs that have been extensively studied, and their potential role in the specificity of particular NDDs, will be discussed below.

### 3.1. Amyloid-β Autoantibodies and Microtubule Protein Tau Autoantibodies in Alzheimer’s Disease

An overproduction of beta-amyloid precursor protein (APP) and accumulation of beta-amyloid protein (β-AP) in the brain are known to be characteristic of AD and related brain alternations [29]. The first evidence for Aβ Aabs was presented in 1993 and showed a pathological significance in AD patients [30]. Subsequently, various studies have presented Aabs in the peripheral circulation of healthy and AD patients, in free form or complex with Aβ-amyloid peptide [31,32,33,34].

Data from studies show that the serum levels of Aβ Aabs in AD patients are lower, linked to a reduced transfer of Aβ Aabs to blood [34,35,36,37]. This leads to the accumulation and deposition of fibrillary aggregates in the CNS.

Nath et al. found that people with AD have higher levels of Aβ42 Aabs in serum than patients with multiple sclerosis (MS) [38]. The data indicate that an immune response (primarily humoral) to beta-amyloid protein (β-AP) can promote neuronal degeneration in AD patients. Du Y et al. have also investigated whether detecting the presence and different titers of naturally appearing anti-beta-amyloid antibodies in the CSF of AD patients and healthy controls is possible [31]. They found that CSF anti-Abeta antibody levels are significantly reduced in AD patients compared with healthy control subjects. Additionally, McMahon et al. (2000) reported that exposure to a low pH could lead to Aabs’ partial denaturation and increased reactivity [39]. In 2020, a random-effect meta-analysis with 1311 AD patients and 1590 healthy individuals was conducted, demonstrating an increase in Aβ IgG in the blood of AD patients [40]. In the same study, CSF Aβ Aab scores in AD individuals versus healthy showed no difference.

In addition to Aβ Aabs in AD, tau Aabs have also been found in the sera and CSF of AD patients and healthy subjects [41,42,43,44,45]. The tau protein has many possible phosphorylation sites. If these sites are hyperphosphorylated (p-tau) during CFS inflammation in AD patients, then tangled paired helical and straight filaments are formed, leading to neuronal damage. Increased p-tau levels (hyperphosphorylated tau protein) in the CSF and neurons are critical biomarkers of AD [46,47]. The studies on anti-tau antibodies are not as numerous as those on anti-Aβ antibodies, but tau Aabs have been found in healthy subjects [44,48] and children [45].

Various studies have reported lower or higher levels of tau Aabs in the sera of AD patients and healthy subjects. For example, in a pilot study, Rosenmann et al. reported increased levels of IgM Abs against p-tau in AD patients [49]. In contrast, Bartos et al. observed decreased levels of tau-reactive Aabs in the sera of AD patients compared with controls [41]. In an exploratory study, Klaver et al. tested the IgM and IgG values from AD, MCI (mild cognitive impairment), and control subjects to p-tau and non-p-tau [43]. They reported increased binding of IgG to p-tau in MCI subjects compared to AD patients and healthy controls, and no changes in IgM values to p-tau and non-p-tau.

In contrast with AD patients, anti-tau antibodies in other dementia patients decreased significantly [42]. In 2015, Hromadkova et al. examined pooled IgG fractions from thousands of healthy donors and showed that they contained anti-tau reactive antibodies [48]. Another study investigated different types of sera Aabs via tau deposition in vitro and found that these antibodies recognized tau filaments and blocked aggregation [50].

Pathological protein aggregation and misfolding are critical to the pathophysiology of many neurodegenerative diseases. In Alzheimer’s disease (AD), there is an accumulation of β-amyloid (Aβ) and tau proteins, which form the characteristic for the disease’s senile plaques and neurofibrillary tangles [51]. The soluble components of plaques are the beta-amyloid peptides, which are proteolytic fragments of the transmembrane amyloid precursor protein, and the building blocks of tangles are comprised of tau protein: a brain-specific, axon-enriched, microtubule-associated protein. It has been hypothesized for a long time that amyloid-beta pathology precedes, and is the inducing factor for, tau phosphorylation and aggregation. Still, new data show that they may co-exist and propel each other from the earliest stages of disease [51].

Over recent years, an ever-increasing body of evidence has suggested that soluble forms of Aβ and tau collaborate, regardless of their aggregation into plaques and tangles, to propel intact neurons into a spiral of death [52]. Moreover, the defining neuronal toxic features of Aβ necessitate the presence of tau. For example, immediate neuronal death, delayed neuronal death following abnormal cell cycle re-entry, and impaired synaptic function are initiated by soluble Aβ types extracellularly and rely on the availability of soluble tau within the cell cytoplasm [52].

In the so-called Lewy body spectrum disorders—Parkinson’s disease, Parkinson’s disease with dementia, and dementia with Lewy bodies—there is an accumulation of insoluble aggregates of a different protein, α-synuclein (α-syn) [53].

Historically, the pathologic differentiation between Alzheimer’s disease and Lewy body spectrum disorders has been based on the identification of extracellular amyloid-beta senile plaques and intracellular neurofibrillary tangles in AD versus Lewy bodies and Lewy neurites in Lewy body spectrum disorders [54]. However, it has been found that, on post-mortem examination, more than half of AD patients show significant Lewy body pathology. Not only do those findings represent a novel view of the pathology of well-defined neurodegenerative diseases, but they also showed a tendency of this subpopulation—called the Lewy body variant of Alzheimer’s disease (AD-LBV)—to exhibit worse rates of cognitive decline and lower overall survival compared to “pure” AD patients [54]. Moreover, beta-amyloid plaques have been identified in patients who have dementia with Lewy bodies [55].

Furthermore, the transactive response DNA-binding protein of 43 kDa (TDP-43) is an intranuclear protein, and cytoplasmic inclusion bodies containing phosphorylated and truncated forms of TDP-43 are classically identified in amyotrophic lateral sclerosis (ALS) [56]. In addition, TDP-43 inclusion bodies are found in more than half of AD patients, most often in the structures of the limbic system and, similarly to the co-deposition of alpha-synuclein, lead to more rapid rates of cognitive decline when compared to “pure” AD patients. Furthermore, an association has been found between the most common genetic risk factor for AD—apolipoprotein E4 (APOE4)—and increased frequency of TDP-43 pathology [56].

Despite many studies supporting the significance of Aβ and tau Aabs and their changes in the sera and CSF of AD and MCI patients, the results are controversial and require more extensive research. There is a lack of consensus on whether these Aabs have any protective role (especially tau Aabs) and whether they could influence the disease spread and cognitive functions.

### 3.2. α-Synuclein Autoantibodies in Parkinson’s Disease

Alpha-synuclein (α-syn) is a presynaptic protein expressed in various brain regions, nerve cells, serum, CSF, plasma, and hematopoietic cells [57,58,59,60,61]. In neurons, α-syn is involved in dopamine synthesis, neurotransmitter release and synaptic plasticity, synaptic vesicle transport and maintenance of recycling-pool homeostasis, lipid metabolism, and inducing innate and adaptive immunity [62,63,64,65,66]. When the production and aggregation of α-syn are disturbed, microglial cells are activated, and the mass secretion of cytokines such as IL-1β, TNF-α, and IL-6 begins. This is linked to brain atrophy, neurodegeneration, and impairment to cognitive functions [67,68].

Gathering neuronal α-syn aggregates is a central process in the pathogenesis of Parkinson’s disease (PD) [69]. These aggregates are found in the substantia nigra pars compacta in PD patients but can also be found in different neurons in the CNS and peripheral nervous system (PNS) [70]. The most extensive and highly structured forms of α-syn aggregate are Lewy bodies, either dense or diffuse structures [71,72].

Similar to Aβ and tau proteins, local immune mechanisms in the brain are observed when there is a pathological increase in α-syn. For example, in PD patients, α-syn aggregates in the substantia nigra co-localize with IgG, indicating that a local Aab immune response is induced [73]. The analysis of α-syn Aabs has varied and is controversial, but different studies have been conducted. Some reports have found that α-syn Aab levels remain unchanged [74,75,76], while others have found that levels are increased [77,78,79] or decreased [80,81]. For example, studies of α-syn Aabs in PD sera and plasma reported higher levels of α-syn Aabs in PD patients than in healthy subjects [77,78]. Besong-Agbo et al. said that the levels of α-syn Aabs in PD sera were lower than in healthy controls and AD patients [80]. Another small study investigated the levels of α-syn Aabs in the sera of PD patients and healthy subjects [75]. They divided the patients according to the duration of PD disease (one group with less than 5 years of PD and a second group with more than 10 years of PD). They reported elevated serum levels of α-syn Aabs in both groups of PD patients compared to healthy subjects. The data also show that the longer the disease process, the more the activity of the antibodies gradually decreases over time [82]. A study separated PD patients and controls by gender, showing that the level of α-syn Aabs in PD patients and healthy male titers was higher than in women [83].

Several studies have also investigated α-syn changes in CSF. Some found increased Aab levels in CSF in PD patients [78,83], while others reported no differences from healthy subjects [74,81]. Concentrations of α-syn Aabs were increased in dementia with Lewy bodies (DLB) and slightly less elevated in AD patients [84].

All these data indicate that the immune response against α-syn in some NDDs can be influenced by many factors, such as sex, age, stage of the disease, severity, and others. It has been proposed that these Aabs may also help in diagnosing PD and explaining its clinical heterogeneity.

### 3.3. Anti-Myelin Basic Protein (-MBP) and Anti-Myelin Oligodendrocyte Glycoprotein (MOG) Autoantibodies in Multiple Sclerosis

MS is an autoimmune, demyelinating NDD of the CNS with an unknown pathogenesis and etiology. The immune system attacks the brain and/or spinal cord, and progressive white and gray matter degeneration is a part of the disease [85,86].

Anti-myelin basic protein (-MBP) Aabs have been found to play a pathogenic role in MS [87]. Such Aabs against MBP have been detected in serum [88,89,90] and CSF [91,92,93] in MS patients. Anti-myelin antibodies may participate in demyelination in MS by activating the complement system, initiating complement-mediated tissue damage [94,95,96,97].

Some studies used healthy subjects as controls and assessed the levels of anti-MBP antibodies or MBP-immune complexes in normal sera [88,98,99]. Overall, the general conclusion is that they are absent from the sera of healthy individuals but are found in the sera of MS patients. However, Hedegaard et al. showed that sera from MS patients contained MBP-reactive antibodies, and their levels were not significantly different from the healthy controls [100]. The antibodies in sera from healthy subjects may be considered natural Aabs, which are harmless [101].

MBP has long been investigated in the pathogenesis of some autoimmune diseases and NDDs, such as MS. MS is characterized by neuroinflammation, demyelination, and axonal loss and impairments that could be linked to an anti-MBP immune response, as well as the presence of anti-myelin antibodies in CSF and serum. This makes it possible to discuss the possibility of using these anti-myelin Abs as biomarkers for MS.

The myelin oligodendrocyte glycoprotein (MOG) is a surface protein produced by the oligodendrocytes in the CNS that has a crucial role in maintaining the stability and functioning of myelin sheaths [102]. Research in recent decades has proven that autoimmunity against MOG exists, mimicking the clinical course of multiple sclerosis and neuromyelitis optica (NMO). In fact, antibodies against MOG were first discovered in NMO patients that were seronegative for the aquaporin-4 IgG antibodies characteristic of the disease [102].

A new term has been introduced because of the variety of clinical presentations that anti-MOG antibodies can exhibit—MOG antibody disease (MOGAD). It encompasses a wide range of inflammatory demyelinating disorders of the CNS—acute disseminated encephalomyelitis (ADEM), brainstem or cortical encephalitis, unilateral or bilateral optic neuritis, chronic relapsing inflammatory optic neuropathy, transverse myelitis, and aquaporin-4 IgG-negative neuromyelitis optica spectrum disorders [103].

### 3.4. Neurofilament Autoantibodies in AD, PD, MS, and ALS

Neurofilaments (NFs) are found in neurons, especially in large myelinated axons from the adult CNS and PNS. There are six types, and following their synthesis and assembly, NFs are transported along the axons [104,105]. Neurons were found to be enriched in pan-neuronal type IV NFs, which are light (NF-L), middle (NF-M), heavy (NF-H), α-internexin, and peripherin [106,107].

Moreover, neurofilaments are very important in the growth and stability of axons [108], for the microtubule content [109], for the preservation of dendritic spine structure and function, and in the regulation of glutamatergic and dopaminergic synapses [110]. Their frequent disorganization impairs synaptic plasticity, memory formation, and behavior, observed in different neuropathologies [111].

In line with this, several studies in NDDs have reported elevated blood levels of NF in the case of neurological damage [112,113,114]. Accordingly, finding NF in serum and CSF in NDD patients may mislead the diagnosis. However, their elevation and the presence of reactive Abs show a very high possibility of being a precise means of detecting the initial phase and neurodegeneration. Elevated levels of NF in blood and CSF are known to suggest axonal damage, which can result from both brain aging and pathological processes [107].

Like Aabs described above, NF Aabs can be detected in the serum and CSF in diseased and healthy individuals. In 2004, Ehling et al. reported higher sera NF-Aab levels in MS patients than in controls [115]. In 2013, Fialova et al. used the serum and CSF levels of anti-NF Aab to follow the MS progression [101]. Other studies have documented that NF concentrations in CSF and peripheral blood are elevated in individuals with MS [116,117,118,119,120,121,122].

Several studies have shown that NF-Aabs could exacerbate the disease in AD patients and some other NDDs [123,124,125]. Bartos et al. showed that the serum levels of NF-H Aabs in AD patients were lower than in healthy subjects [41]. The study by Soussan et al. also compares NF-Aab profiles in the serum of healthy individuals and AD patients [126]. An equal binding of different NF-H isoforms (ventral and dorsal root) was reported in controls, whereas in AD, the levels of Aabs against ventral NF-H were higher than those against dorsal NF-H.

Additionally, NF-Aabs were found using the immunofluorescence technique in serum samples from PD patients [127]. People over 70 years, with or without disease, have significantly higher levels of antibodies than younger age groups. In addition, two other studies also revealed increased levels of Aabs in PD [128,129,130].

Similarly, the ELISA method detected NF Aabs in serum from ALS patients. ALS patients have higher levels than healthy individuals and those with other diseases [131]. Several more studies confirm elevated NF-Aabs in CSF or serum in ALS patients [131,132,133,134].

Since the neurofilament network is disturbed in different neurological diseases, and neurofilaments are released into the peripheral blood and CSF, they can be detected in these body fluids. Neurofilaments are highly immunogenic, induce a specific antibody response, and release Aabs. The levels of NF-Aabs correlate with the onset of various neurological diseases. Studying their role in the pathogenesis of the conditions and their use in diagnosing NDDs can be very useful in clinical practice.

A schematic presentation of the described possible mechanisms of autoimmune response to pathology-related molecules of the most common neurodegenerative diseases is shown in Figure 1.

There are also other antibodies presumed to be related to NDDs, such as anti-elastin antibodies. However, the significance of elastin-derived peptides and their role in NDDs, and the usefulness of anti-elastin antibodies, remain inconclusive [135].

## 4. Autoantibodies as Biomarkers for Neurodegenerative Disease Diagnosis

Since NDDs are debilitating diseases that limit the life span and quality of life of suffering patients, much effort has been made to find reliable biomarkers. The ideal biomarkers would be helpful for diagnosis, prognosis, and managing patients’ therapeutic response and follow up. Finding such markers, ideally based on the pathobiology of the disease, would benefit the application of precision medicine [11]. Some Aabs could be pathogenetic, and some could participate in debris clearance. However, new translational strategies based on “cause and/or effect” are assumed to be needed before the full adoption of clinical practical biomarkers [136].

We must state that some Aabs against specific molecules (i.e., NDD pathophysiology-related molecules, etc.) could help in diagnosis. However, some pathology-related molecules (e.g., neurofilaments) also could be used as biomarkers.

For example, amyloid proteins and their fragments could be employed for diagnosis, patient stratification, and follow up. However, due to their presence and accumulation in the brain and CSF, their use is limited to invasive procedures with increased health risks [11]. In addition to the role of Aabs in NDD diagnosis and management, some new biomarkers are on the horizon. Gafson et al. discussed the neurofilament proteins as potential biomarkers involved in the pathogenesis of the diseases and, therefore, probably suited to serve as markers for neurodegeneration [106].

Similar outcomes were reported by Khalil et al. in 2018. They discussed the role of neurofilaments in neuroaxonal damage, which could be assessed both in blood and CSF. Recent research connected neurofilaments to MS, neurodegenerative dementia, stroke, traumatic brain injury, PD disease, and ALS [107].

Additionally, Hansson et al. used a blood-based assessment of neurofilaments to distinguish PD from other disorders, which are atypical Parkinsonian disorders. They confirmed that blood testing is enough for the differential diagnosis of these entities [114].

Zetterberg and Burnham also focused on the blood-based biomarkers for AD, such as plasma Aβ, plasma tau, plasma neurofilament light, and panels of biomarkers (i.e., protein biomarkers panel, panels of blood-based biomarker-associated disease phenotypes, metabolomics, miRNA biomarker panels, exosomes, etc.) [112].

Usually, brain-derived biomarkers are at extremely low concentrations in the blood, which is one of the challenges when assessing the utility of these biomarkers. However, CSF is a preferable sample because the biomarkers are at their actual concentration, and there are few heterophilic antibodies [112]. We must also bear in mind that heterophilic antibodies unrelated to NDDs could be found in serum and may falsify the results. In addition, it is vital to include Aabs in a blood-based screening/triage algorithm for primary care settings to evaluate possible NDDs early, thus identifying useful Aabs in serum and performing further testing using CSF samples and PET [112].

Similar to these results are the investigations of Yin and Stover, who introduced Aabs for AD diagnosing and monitoring, considering the qualitative changes in the levels of these antibodies with disease progression: amyloid-β precursor proteins, τ protein, S100b, and phospholipid [137].

In addition, the most recent study in the field was conducted by Gao et al., who investigated the functional and pathological effects of α-synuclein on synaptic SNARE (Snap receptor) complexes in PDD. However, this biomarker has not been studied enough and could not be transferred directly to clinical practice [138].

Much more practical is the testing of Aabs for NDDs—they are non-invasive, usually low-cost, and available in CSF and peripheral blood. Antibodies to different neuronal proteins are promising biomarkers, studied in clinical trials [12,136,139,140,141,142,143].

One of the most desirable functions of Aabs as biomarkers is to predict the disease. According to Lesie et al., since Aabs are produced after T and B cell activation, Aabs could be a marker for pathological process development and predict the likelihood of clinical symptom development and disease progression [144]. The authors suggested that if Aabs could measure the disease severity, theoretically, they could help predict disease, especially for disorders with long preclinical periods and multiorgan involvement. Furthermore, if the biomarkers could indicate the onset of the disease, one could assume that the disease could be prevented. Additionally, accurate prediction could help avoid secondary and tertiary complications of the already-diagnosed illness [144].

However, assessing the sensitivity, specificity, and positive predictive values (PPVs) of the prediction is essential. The PPV is obtained via dividing the number of Aab-positive patients who acquire clinical illness by the total number of Aab-positive subjects. Biomarkers’ prognostic relevance differs in groups with varying levels of risk of developing the disease. When the illness risk is high, the prediction power is high, but when the risk of developing disease is low, like in the general population, the predictive power is reduced [145]. However, cross-sectional predictions must be validated in prospective investigations. Researchers rely on cross-sectional data from cases with established clinical illnesses to provide prediction values. This technique is generally faulty and pertains to illness case identification rather than prognosis [140].

Although, despite intensive research in the field, establishing biomarkers for NDDs is still challenging, the hope is that testing Aabs to neuronal antigens could provide specific, rapid, and affordable means of diagnosing these diseases. However, when Aabs are produced locally, and the BBB is preserved, these antibodies would not be in sufficient concentrations in the peripheral blood (sometimes, a colossal discrepancy between blood vs. CFS levels was found). This informs future research in improving sampling and developing more sensitive commercially available kits for detection in serum samples.

Another challenge arises when a considerable number of specific Aabs are found in the blood of healthy subjects, which calls into question the specificity and precision of these tests. Also, the research has shifted from discussing the Aabs as causing or exacerbating disease to viewing them as biomarkers for clinical diagnosing [12].

In some cases, NDDs and autoimmune encephalitis have common pathophysiological and clinical features; thus, antibodies to neuronal surface antigens, for example, could be linked to neurodegeneration. This makes distinguishing between the various conditions based on the presence of Aabs challenging [10].

There is also a need for more studies assessing the clinical usefulness of Aabs, their origin, their mechanisms of production, and their physiological presence (in healthy subjects). Answering these questions will give us the confidence to use certain Aabs in NDDs, interpret them, and use them for clinical purposes.

After many years of unfortunate outcomes in clinical trials, new monoclonal antibodies (MAbs) have redefined the therapeutic range for Alzheimer’s disease, and more such treatments are currently being clinically researched. As of 2023, the FDA approved two anti-amyloid monoclonal antibodies—aducanumab and lecanemab. Those two antibodies represent the first disease-modifying treatments for AD, as well as being among the first disease-modifying treatments for any neurodegenerative disease [146].

In its Phase 3 trial, lecanemab showed significant reductions in cognitive decline as measured by the Alzheimer’s Disease Assessment Scale’s cognitive subscale (ADAS-cog) and the Alzheimer’s Disease Cooperative Study (ADCS) Activities of Daily Living (ADL) measure. Simulation modeling suggests that lecanemab treatment may extend the mild phases of Alzheimer’s disease dementia by approximately 2.5 years, offering substantial cost savings and addressing patients’ and families’ priorities of delaying disease progression, maintaining autonomy, and avoiding increased dependency. It is anticipated that future disease-modifying therapies will permanently alter the course of Alzheimer’s disease [147].

These unprecedented groundbreaking therapies come with new demands for the various stakeholders involved in AD and the care of our elderly population. Anti-amyloid MAbs represent a milestone to be celebrated, as they establish a new era for addressing the challenges posed to our most valuable global asset—the human brain [146].

Currently, there is no known disease-modifying drug for the treatment of Parkinson’s disease and other alpha-synucleinopathies. A recombinant humanized anti-alpha-synuclein IgG1 monoclonal antibody, Cinpanemab, targeted against aggregated α-synuclein, is currently in Phase 2 trial [148]. Another high-affinity alpha-synuclein monoclonal antibody, called MEDI1341, binds both monomeric and aggregated forms and can sequester extracellular alpha-synuclein and decrease its spreading in vivo. This MAb is now in a Phase 1 clinical trial, bringing hopes of finding a progression-modifying treatment for PD and probably also other synucleopathies [148].

## 5. Conclusions

NDDs impact millions globally, arising from protein misfolding, inflammation, and cellular abnormalities. Emerging evidence shows Aabs contributing to demyelination, cognitive decline, and altering NDD understanding. Immune system involvement, specific Aabs against NDD-related molecules, and their potential as diagnostic biomarkers are explored. NDD autoimmune aspects could enhance early detection and reveal options for novel treatment.

## Figures and Tables

**Figure 1 antibodies-12-00081-f001:**
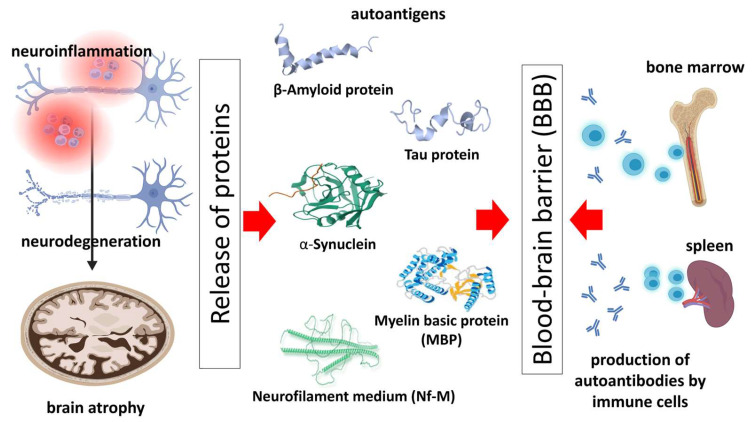
Possible mechanisms of autoimmune response to pathology-related molecules of the most common neurodegenerative diseases. Neuroinflammation and neurodegeneration eventually lead to brain atrophy, accompanied by the production and release of pathologic proteins. The latter serve as autoantigens for producing autoantibodies via bone marrow, spleen immune cells, etc. The blood–brain barrier is essential in the process, since the disrupted barrier could allow the transportation of autoantibodies from the bloodstream to the nervous system and vice versa.

## Data Availability

Not applicable.

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
