# Peer review of "Role of Specific Autoantibodies in Neurodegenerative Diseases: Pathogenic Antibodies or Promising Biomarkers for Diagnosis"

_2073-4468, 2023, doi:10.3390/antib12040081_

Round 1

Reviewer 1 Report

Comments and Suggestions for Authors

Authors presented the review paper on the presences of autoantibodies against the pathological proteins in neurodegenerative diseases, which is an important topic with great interest to many researchers in the field.

However, authors need to be more detail on the presences of autoantibodies in each protein with specific percentages, epitopes and trends in each specific patients.

Recently, patients with double pathologies are being identified, such as amyloid-beta and tau, amyloid-beta and alpha-synuclein, amyloid-beta and TDP-43, etc...

Authors need to address the locations of autoantibodies in our body system and the pathologies of each disease.

Comments on the Quality of English Language

Authors need to improve the transitions between paragraphs.

Author Response

Reviewer's Comment

Author's Response

Authors presented the review paper on the presence of autoantibodies against pathological proteins in neurodegenerative diseases, which is an important topic with great interest to many researchers in the field.

We genuinely appreciate the reviewer's recognition of the importance of our paper. It's encouraging to know that our topic aligns with the interests of fellow researchers.

However, authors need to provide more detail on the presence of autoantibodies in each protein with specific percentages, epitopes, and trends in each specific patient.

We thank the reviewer for their suggestion to enhance the paper's depth. In our revised manuscript, we included specific and more detailed information of the antibodies, when we found such data.

Recently, patients with double pathologies are being identified, such as amyloid-beta and tau, amyloid-beta and alpha-synuclein, amyloid-beta and TDP-43, etc.

The identification of patients with double pathologies is indeed an emerging area of interest. We addressed this aspect more comprehensively in our revised manuscript, incorporating relevant studies and exploring the implications of these co-occurrences.

Authors need to address the locations of autoantibodies in our body system and the pathologies of each disease.

We acknowledge the need to elucidate the locations of autoantibodies within the body system and their relationships with disease pathologies. In the revised manuscript, we provided a more comprehensive analysis of these factors, when applicable.

Authors need to improve the transitions between paragraphs.

Ensuring smoother transitions between paragraphs is essential for a seamless reading experience. We appreciate the suggestion and enhanced the overall logical flow of the paper in the revised version.

Reviewer 2 Report

Comments and Suggestions for Authors

The review is well structured and well written. It covers many concepts to be considered about the autoimmune nature of neurodegenerative diseases and, more specifically, the role of some antibodies. However, we believe that this review could capture the attention of potential readers more if it included some ideas:

In section 3.3. it would be highly recommended to mention a protein well known for its antigenic character and its role in the development of multiple sclerosis. It is MOG (Myelin oligodendrocyte glycoprotein). In fact, this protein is used to induce in animal models the disease equivalent to multiple sclerosis (experimental autoimmune encephalitis, EAE).

When the authors start talking about cerebrospinal fluid (CSF) they should comment on the connotations of antigens and antibodies that are able to cross the blood-brain barrier.

Within these ideas, it would be very interesting to comment on something that is used in European countries for the clinical diagnosis of patients with multiple sclerosis. These are oligoclonal bands (OCB), since they are, after all, antibodies (which is what this review is about).

It would also be very useful to make a brief comment on the monoclonal antibodies used in patients with neurodegenerative diseases for therapeutic use (since in many cases they are used as the only treatment).

Globally, two ideas related to the autoimmune nature of neurodegenerative diseases should also be discussed (albeit briefly):

(i) The relationships between vitamin D and neurodegenerative diseases (which has been known for several decades). This is of interest in this review because it is related to the anti-inflammatory and immunomodulatory character of vitamin D.

ii) the relationships between gut microbiota and neurodegenerative diseases. This is of interest because many theories point to the immune reaction to the microbiota as an important etiological factor. For this reason, for several years now we no longer speak of the gut-brain axis but of the microbiota-gut-brain axis (especially in neurodegenerative diseases).

It would also be very useful to make a brief comment on the monoclonal antibodies used in patients with neurodegenerative diseases for therapeutic use (since in many cases they are used as the only treatment).

Globally, two ideas related to the autoimmune nature of neurodegenerative diseases should also be discussed (albeit briefly):

(i) The relationships between vitamin D and neurodegenerative diseases (which has been known for several decades). This is of interest in this review because it is related to the anti-inflammatory and immunomodulatory character of vitamin D.

ii) The relationships between gut microbiota and neurodegenerative diseases. This is of interest because many theories point to the immune reaction to the microbiota as an important etiological factor. For this reason, for several years now we no longer speak of the gut-brain axis but of the microbiota-gut-brain axis (especially in neurodegenerative diseases).

Author Response

Reviewer's Comment

Author's Response

The review is well structured and well written. It covers many concepts to be considered about the autoimmune nature of neurodegenerative diseases and, more specifically, the role of some antibodies. However, we believe that this review could capture the attention of potential readers more if it included some ideas:

We appreciate the positive feedback on the overall structure and writing quality of our review. It's heartening to know that the reviewer finds the topic relevant.

In section 3.3, it would be highly recommended to mention a protein well known for its antigenic character and its role in the development of multiple sclerosis. It is MOG (Myelin oligodendrocyte glycoprotein). In fact, this protein is used to induce in animal models the disease equivalent to multiple sclerosis (experimental autoimmune encephalitis, EAE).

Thank you for this valuable suggestion. We acknowledge the significance of MOG in the context of multiple sclerosis and its relevance to the autoimmune nature of neurodegenerative diseases. In our revised manuscript, we incorporated a section discussing the role of MOG and its implications.

When the authors start talking about cerebrospinal fluid (CSF), they should comment on the connotations of antigens and antibodies that are able to cross the blood-brain barrier.

We recognize the importance of elucidating the significance of antigens and antibodies capable of crossing the blood-brain barrier in the context of neurodegenerative diseases. In our revised manuscript, we expanded upon this aspect to provide a comprehensive understanding for our readers.

It would also be very useful to make a brief comment on the monoclonal antibodies used in patients with neurodegenerative diseases for therapeutic use (since in many cases they are used as the only treatment).

We appreciate this suggestion and agree on the importance of discussing monoclonal antibodies used for therapeutic purposes in neurodegenerative diseases. In the revised manuscript, we included a section providing insights into the therapeutic use of monoclonal antibodies.

Globally, two ideas related to the autoimmune nature of neurodegenerative diseases should also be discussed (albeit briefly): (i) The relationships between vitamin D and neurodegenerative diseases (which has been known for several decades). This is of interest in this review because it is related to the anti-inflammatory and immunomodulatory character of vitamin D. ii) The relationships between gut microbiota and neurodegenerative diseases. This is of interest because many theories point to the immune reaction to the microbiota as an important etiological factor. For this reason, for several years now we no longer speak of the gut-brain axis but of the microbiota-gut-brain axis (especially in neurodegenerative diseases).

We express our gratitude for these excellent suggestions. We concur that exploring the relationships between vitamin D, gut microbiota, and their roles in neurodegenerative diseases aligns well with the scope of our paper. In our revised manuscript, we incorporated sections discussing these vital aspects, thus enhancing the comprehensiveness of our review.

Reviewer 3 Report

Comments and Suggestions for Authors

The manuscript “Role of specific autoantibodies in neurodegenerative diseases: Pathogenic antibodies or promising biomarkers for diagnosis” (Manuscript ID: antibodies-2615339) refers to an interesting and challenging research topic. The authors have managed to provide a great number of references, recent and older ones, concerning the presence of autoantibodies targeting specific pathology-related molecules/entities, such as Aβ and α-synuclein -as well as myelin basic protein and neurofilaments, in individuals with various neurodegenerative diseases, including AD, PD, MS and ALS. A weak point of the manuscript is the rather obscure final Part 4, i.e., “The potential of autoantibodies to be biomarkers for neurodegenerative diseases diagnosis”.  More specifically:

In Part 4, the authors should clearly distinguish between the possible diagnostic role of i) autoantibodies against specific pathology-related molecules/entities and ii) pathology-related molecules/entities (e.g., NFs) per se.

Some lines in Part 4 are not quite clear, e.g., lines 337-342:Although brain-derived biomarkers could be at extremely low concentrations in the blood, there may be heterophilic antibodies, which may falsify the results (both to increase or decrease the actual levels falsely). However, CSF is a preferable sample not only because the biomarkers are at their actual concentration but also because there are not so many heterophilic antibodies [96]. However, interpretation of neurofilament measurement should be made with caution, considering all the discussed above Aabs. Do the authors mean that possible co-existence of autoantibodies with the corresponding pathology-related molecule/entity in the same clinical sample might affect e.g., the apparent concentration of the latter?? Also, the meaning of lines 342-345 (“In addition, it is vital to develop a blood-based screening/triage algorithm for the primary care settings to evaluate possible NDD early, thus identifying them to perform further testing using CSF samples and PET [96]”) is obscure.

As mentioned in Part 3, lines (143-145), “Different Aabs can be assessed in the cerebrospinal fluid (CSF) and blood (serum) of patients with NDDs. Their concentration in the blood is significantly higher than in the CSF”. On the other hand, in Part 4 (lines 378-382) it is stated that “However, some of the Aabs are not in sufficient concentrations in the peripheral blood (sometimes, a colossal discrepancy between blood vs. CFS levels was found).” Since there seems to be some inconsistency between the two statements, at least in my opinion, the authors should further discuss and better clarify.

Moreover:

The manuscript includes one Figure, which could be more descriptive/informative. In addition, the Figure legend should provide more explanatory details.  For instance, the “autoantigens” mentioned in the Figure should be clearly specified (e.g.,  as the “proteins released” due to the neurodegeneration?); moreover, the “immune cells produced” and the (auto)antibodies generated should be clearly labeled as such in the Figure and mentioned in the Figure legend.

Minor comments:

Line 168: 2000 instead of 2007; line 243: Aabs instead of aAbs.

Comments on the Quality of English Language

Minor editing is required.

Author Response

Reviewer's Comment

Response

The manuscript “Role of specific autoantibodies in neurodegenerative diseases: Pathogenic antibodies or promising biomarkers for diagnosis” (Manuscript ID: antibodies-2615339) refers to an interesting and challenging research topic. The authors have managed to provide a great number of references, recent and older ones, concerning the presence of autoantibodies targeting specific pathology-related molecules/entities, such as Aβ and α-synuclein -as well as myelin basic protein and neurofilaments, in individuals with various neurodegenerative diseases, including AD, PD, MS and ALS. A weak point of the manuscript is the rather obscure final Part 4, i.e., “The potential of autoantibodies to be biomarkers for neurodegenerative diseases diagnosis”.

Thank you for your insightful review and for recognizing the importance of our paper. We value your constructive feedback and addressed your concerns to improve the manuscript further.

In Part 4, the authors should clearly distinguish between the possible diagnostic role of i) autoantibodies against specific pathology-related molecules/entities and ii) pathology-related molecules/entities (e.g., NFs) per se.

We appreciate your suggestion and revised Part 4 to make a clear distinction between the diagnostic potential of autoantibodies targeting specific pathology-related molecules/entities and the molecules/entities themselves. We believe this enhanced the clarity of our discussion.

Some lines in Part 4 are not quite clear, e.g., lines 337-342...

We understand your concern and apologize for the ambiguity in those lines. We rephrased this section to better convey the meaning and to enhance clarity.

As mentioned in Part 3, lines (143-145), “Different Aabs can be assessed in the cerebrospinal fluid (CSF) and blood (serum) of patients with NDDs. Their concentration in the blood is significantly higher than in the CSF”. On the other hand, in Part 4 (lines 378-382) it is stated that “However, some of the Aabs are not in sufficient concentrations in the peripheral blood (sometimes, a colossal discrepancy between blood vs. CFS levels was found).” Since there seems to be some inconsistency between the two statements, at least in my opinion, the authors should further discuss and better clarify.

Thank you for pointing out the discrepancy between the statements in Part 3 and Part 4. We addressed this inconsistency by discussing the variations in antibody concentration more comprehensively.

The manuscript includes one Figure, which could be more descriptive/informative. In addition, the Figure legend should provide more explanatory details.  For instance, the “autoantigens” mentioned in the Figure should be clearly specified (e.g.,  as the “proteins released” due to the neurodegeneration?); moreover, the “immune cells produced” and the (auto)antibodies generated should be clearly labeled as such in the Figure and mentioned in the Figure legend.

We appreciate your suggestion regarding the figure. We enlarged the description and provide more explanatory details in the figure legend. We revised the figure to improve clarity and comprehension.

Minor comments: Line 168: 2000 instead of 2007; line 243: Aabs instead of aAbs.

Thank you for catching these minor errors. We corrected them as per your recommendations to ensure the accuracy of the manuscript.

Comments on the Quality of English Language

Minor editing is required.

We appreciate your comments on the quality of English language. We performed editing to enhance the manuscript's readability and coherence. Your feedback is valuable in ensuring the paper meets the highest standards.

Reviewer 4 Report

Comments and Suggestions for Authors

Review of the manuscript entitled: Role of Specific Autoantibodies in Neurodegenerative Diseases: pathogenic antibodies or promising biomarkers for diagnosis.

The manuscript is interesting but some corrections should be made:

1.      The manuscript lacks “aim”. In end of introduction clear aim of the manuscript must be added e.g. "The aim of the present study was to ...".

2.      References are needed: lins 38,

3.      If you introduce abbreviations, please explain when you first use them e.g. line 86 “Aβ”, line 87 “ROS” and so on….

4.      The content of the manuscript is correct, but I would like to ask whether anti-elastin antibodies have any significance in the development of neurodegenerative diseases? This seems to be crucial because the amount of peptides formed after the breakdown of elastin (elastin derived peptides) correlates with the number of neurodeneterative diseases, in particular AD. Maybe it's worth taking this into account?

Author Response

Reviewer's Comment

Response

The manuscript lacks an "aim." In the end of the introduction, a clear aim of the manuscript must be added e.g. "The aim of the present study was to ..."

Thank you for this suggestion. We included a clear and concise statement of the aim of our study at the end of the introduction to provide a more focused and informative introduction.

References are needed: lines 38

We apologize for the missing reference and promptly added the required citation to line 38 to ensure proper attribution.

If you introduce abbreviations, please explain when you first use them e.g. line 86 “Aβ”, line 87 “ROS” and so on…

Thank you for pointing this out. We explained the abbreviations when they are first introduced in the manuscript to improve the clarity and comprehension of the text for our readers.

The content of the manuscript is correct, but I would like to ask whether anti-elastin antibodies have any significance in the development of neurodegenerative diseases? This seems to be crucial because the amount of peptides formed after the breakdown of elastin (elastin-derived peptides) correlates with the number of neurodegenerative diseases, in particular AD. Maybe it's worth taking this into account?

Your suggestion regarding the potential significance of anti-elastin antibodies in neurodegenerative diseases is interesting and relevant. We added a paragraph on this topic at the end of section 3.

Round 2

Reviewer 1 Report

Comments and Suggestions for Authors

Authors have revised the manuscript according to the comments.

Authors need to address the differentiation between neurodegenerative diseased patients with or without autoantibodies, since not all patients with neurodegenerative diseases have autoantibodies.

How can we segregate above patients?

With advances with blood biomarkers, especially with pathological oligomers, authors need to discuss the implementations or comparisons with autoantibodies

Too many run-on sentences need to be improved.

Comments on the Quality of English Language

Too many run-on sentences need to be improved.

Author Response

Reviewer 1

Authors have revised the manuscript according to the comments.

We are thankful for your valuable comments that are essential to enhancing the quality and readability of our paper.

Authors need to address the differentiation between neurodegenerative diseased patients with or without autoantibodies, since not all patients with neurodegenerative diseases have autoantibodies.

How can we segregate above patients?

We appreciate the reviewer's insightful comment. In the revised manuscript, we have included a subsection at the end of point 2. specifically addressing the difficult differentiation between neurodegenerative disease patients with and without autoantibodies. However, it's crucial to note that the involvement of antibodies in neurodegenerative diseases is an active area of research, and our understanding may evolve over time.

With advances with blood biomarkers, especially with pathological oligomers, authors need to discuss the implementations or comparisons with autoantibodies

Thank you for highlighting this important aspect. In the revised manuscript, we have expanded the discussion on antibodies in neurodegeneration with a paragraph on pathological oligomers. We have drawn comparisons with autoantibodies, exploring their respective roles and diagnostic capabilities.

Too many run-on sentences need to be improved.

We acknowledge the concern regarding sentence structure. The manuscript has undergone a thorough language revision, addressing run-on sentences to enhance readability and coherence. We have ensured that the revised manuscript adheres to proper grammar and sentence construction norms.

Reviewer 3 Report

Comments and Suggestions for Authors

The manuscript (ID: antibodies 2615339) has been thoroughly revised and the first round's comments have been adequately addressed. I might just propose to add the word "possible" before the word "Mechanisms" at the beginning of the revised legend of Figure 1.

Comments on the Quality of English Language

Minor editing of English language is required

Author Response

Reviewer 3

Comments and Suggestions for Authors

The manuscript (ID: antibodies 2615339) has been thoroughly revised and the first round's comments have been adequately addressed. I might just propose to add the word "possible" before the word "Mechanisms" at the beginning of the revised legend of Figure 1.

Thank you for the suggestion. In response, we have incorporated the word "possible" before "mechanisms" in the legend of Figure 1 to more accurately convey the nature of the proposed mechanisms.

Comments on the Quality of English Language

Minor editing of English language is required

We appreciate the feedback on language quality. The manuscript has undergone a comprehensive language editing process, addressing minor grammatical and stylistic issues to enhance overall clarity and fluency.